# Strengths, Microstructure and Nanomechanical Properties of Concrete Containing High Volume of Zeolite Powder

**DOI:** 10.3390/ma13184191

**Published:** 2020-09-21

**Authors:** Zhouping Yu, Weijun Yang, Peimin Zhan, Xian Liu, Deng Chen

**Affiliations:** 1School of Civil Engineering, Changsha University of Science and Technology, Changsha 410114, China; yzping0912@gmail.com; 2Yuanpei College, Shaoxing University, Shaoxing 312000, China; dh54321ggyx@gmail.com; 3School of Materials Science and Engineering, Tongji University, Shanghai 200000, China; zhanpeimin@tongji.edu.cn; 4College of Civil Engineering, Suzhou University of Science and Technology, Suzhou 215011, China; chendeng0310@gmail.com

**Keywords:** zeolite powder, strength, microstructure, nanomechanical properties

## Abstract

In order to save resources and reduce the carbon footprint of concrete, the addition of high volumes of supplementary cementitious materials (SCMs) to replace cement is one of the most effective and promising methods. Zeolite powder (ZP), with a high specific surface area, exhibits high pozzolanic reactivity in cement-based materials. This paper investigates the effects of ZP addition used to replace cement at the levels of 20%, 40% and 60% on the strength development and microstructure evolution of concrete, and the nanomechanical properties are analyzed using nanoindentation technique. The results show that the replacement of ZP for cement generally has a dilution effect on the concrete, leading to a detrimental effect on the strength development. However, the 20% ZP replacement for cement slightly enhances the 90-day compressive strength. The pore structure analysis shows that the sample with 20% ZP content has a lower total porosity than the control sample. The hydration of ZP goes against the dilution effect and reduces the total porosity of concrete to compact the microstructure. Nanoindentation investigation of the matrix shows that 20% ZP decreases the content of portlandite but increases the content of high density calcium silicate hydrate (C-S-H). This is beneficial for improving the nanomechanical properties of interface transition zone. However, further increases in the content of ZP (40% and 60%) decrease the total volume of C-S-H and increase the porosity to degrade the microstructure.

## 1. Introduction

Portland cement (PC) production is a process with high carbon dioxide emission and energy consumption [1]. Recent studies have focused on developing a more sustainable way of mitigating the economic and environmental costs of cement production [2]. Incorporation of supplementary cementitious materials (SCMs) in cement products effectively reduces the consumption of clinker [3]. The main components of aluminosilicate present in SCMs react with the portlandite (CH) in the cement matrix to increase the hydration degree [3]. The appropriate selection of SCMs can enhance the strength development, compact the pore structure and improve the durability of cement-based materials [4,5,6,7,8,9].

Various types of SCMs, including industrial byproducts of fly ash, blast furnace slag and other metallurgical slags, have been used in concrete production for many years [10,11,12,13]. Although the industrial byproducts are commonly regarded as promising green SCMs, the inconsistent property of industrial byproducts is still one of the critical issues to be faced in large-scale deployment [14]. Recently, the production of these industrial byproducts has been limited due to the development of industrial technology and stricter environmental protection policy [5,15]. For example, the restriction in the use of coal in China reduces the availability of fly ash which has been recognized as a class of high quality SCMs. Therefore, it underlines the need for further exploring the use of natural pozzolans, such as zeolite powders (ZP), as SCMs in cement products.

ZP is one of the most abundant deposits occurring in areas of recent and ancient volcanism [16]. The framework in ZP consists mainly of corner-shearing aluminosilicate tetrahedra and exhibits microporous structure [16]. The high specific surface area of the porous zeolite structure enables higher pozzolanic reactivity than the natural pumice and tephra, and other industrial byproducts of blast furnace slag and fly ash [17,18,19]. Uzal and Turanlı [20] observed the 55% reaction degree of ZP in the cement paste after 28 days of hydration. The ZP addition also contributes to the strength development of cement pastes [21]. Snellings et al. [16] investigated the pozzolanic reactions of various natural ZP in the cement pastes by time-resolved synchrotron X-ray powder diffraction. It was observed that ZP addition accelerated the hydration of C_3_S and formations of portlandite and ettringite, and the pozzolanic reactivity of ZP depended on the exchangeable cation content and crystallinity [16]. Sedić et al. incorporated 20% ZP into cement pastes used for geological CO_2_ sequestration to improve carbonation resistance, and observed that ZP effectively compacted the pore structure [22]. Xu et al. [23] also observed that 5%–10% ZP addition could improve the flowability and cohesiveness of cement mortar. It can be found that a considerable amount of work has been published in the literature on cement-based materials containing ZP. However, there are no reports on the properties and microstructure of concrete containing high volume of ZP, and therefore this needs to be investigated further.

Nowadays, the development and application of nanomechanical characterization methods, such as nanoindentation and nanoscratch, show great potential for a better understanding of cement-based materials at the micro-, nano- and even smaller scales, which exhibits direct correlations to the macroscopic mechanical properties of cement-based materials. Some systematic investigations using the nanoindentation technique in cement-based materials have been previously published by Hu et al. [24], Lee et al. [25] and Luo et al. [26]. He et al. [27,28] investigated the effects of SCMs, such as lithium slag and glass powder, on the nanomechanical properties of concrete. It was observed that the incorporation of appropriate content of SCMs significantly enhanced the elastic modulus of interfacial transition zone (ITZ) and high-density calcium silicate hydrate. However, knowledge on the nanomechanical properties of concrete containing ZP is limited. Published literature contains no reports on the nanomechanical properties of concrete containing a high volume of ZP, more than 50% by weight. The aim of this study is to fulfill this need and thus promote the potential application of ZP in concrete.

In this paper, the compressive strength development of concretes containing different amounts of ZP was measured. Meanwhile, the effects of ZP on the microstructure of concrete were tested by using scanning electron microscopy (SEM) and mercury intrusion porosimetry (MIP) methods, and the nanomechanical properties were analyzed using the nanoindentation technique.

## 2. Experimental Section

### 2.1. Raw Materials

The chemical compositions of CEM II 42.5 PC and ZP as determined using an X-ray fluorescence (Thermo Fisher Scientific, Waltham, MA, USA) method are shown in Table 1. ZP is mainly comprised of SiO_2_ and Al_2_O_3_. Both PC and ZP particles have irregular polygonal shapes (Figure 1). The mean particle sizes of PC and ZP, as measured by using laser particle analyzer (Malvern Panalytical, Malvern, UK), were 14.3 μm and 6.5 μm, respectively (Figure 2). River sand with a maximum size of 2.12 mm was used as the fine aggregate. Crushed granite with a size range of 4–18 mm was used as the coarse aggregate.

### 2.2. Mixture Proportions

ZP was added into concrete as a cement replacement in this study. The content of ZP was in the range of 0–60% by weight of cement. The water to binder ratio and binder to aggregate ratio of each group were kept as constants with the values of 0.30 and 0.40, respectively. The content of polycarboxylic superplasticizer was adjusted to maintain a slump of 150–200 mm. The proportions of the concrete mixtures are shown in Table 2. Concretes were casted in 150 mm × 150 mm × 150 mm cubic molds for testing the compressive strength. The specimens were cured in a standard curing room with 20 ± 1 °C and >95% relative humidity. After 24 h of curing, the concretes were demolded and further cured in the 20 °C water until tested.

### 2.3. Test Methods

The compressive strengths of concretes after 7, 28 and 90 days of curing were measured using an auto-test compression machine (KASON, Jinan, China), as shown in Figure 3.

The fractured samples without coarse aggregate were collected. The core parts of fractured samples were coated with gold and the morphologies of cross section were analyzed by using a Quanta FEG 250 field emission scanning electron microscope (SEM) (FEI). The granular samples with 5.0 ± 0.5 mm diameter were collected for characterizing the pore structure by using the mercury intrusion porosimetry (MIP) (Quantachrome, Boynton Beach, FL, USA) with PoreMaster 60. The detailed steps of the above experiments are described in Reference [27].

The nanoindentation technique was carried out to determine the nanomechanical properties of concrete samples containing ZP. To meet nanoindentation measurement requirements [29], the concrete samples were sliced, dried, impregnated in epoxy and carefully polished to achieve a smooth surface, as shown in Figure 4. Figure 5 shows the instrument (Bruker, Billerica, MA, USA) for measuring nanoindentation. Three representative areas on the cement matrix and one representative area on the ITZ of the concrete sample were characterized using nanoindentation. The distance between different indents in the samples was set to 5 μm in the lateral and vertical directions. The indenter came into contact with the sample surface with a trapezoidal holding time of 20 s, a holding time of 20 s at a maximum load of 2 mN and an unloading time of 20 s. The histograms of elastic modulus were determined with the statistical deconvolution [28]. Elastic modulus E was calculated by Equation (1):(1)$$E=\left(1-v^{2}\right)\times {\left[\frac{1}{E_{r}}-\frac{\left(1-v_{i}^{2}\right)}{E_{i}}\right]}^{-1}$$
where *E_i_* = 1140 GPa and *v_i_* = 0.07 are Young’s modulus and Poisson’s ratio of the indenter used in this experiment. *E_r_* is the reduced elastic modulus of the sample and *v* is the Poisson’s ratio of sample [29].

## 3. Results and Discussion

### 3.1. Compressive Strength

Figure 6 shows the effects of ZP addition on the compressive strength development of concrete samples. The 7-day, 28-day and 90-day compressive strengths were in the ranges of 16.5–37.1, 29.6–52.7 and 50.8–66.4 MPa, respectively. ZP had lower reactivity than cement clinker, so the replacement of ZP for cement led to a dilution effect on the concrete. The concretes with ZP addition generally exhibited lower compressive strengths than concretes with 100% PC. It is interesting to note that the replacement of 20% ZP for cement slightly enhanced the 90-day compressive strength. This should be considered from two perspectives: first, the pozzolanic reaction of ZP particles in the concrete promotes the hydration after a long-term curing age; second, ZP has smaller particle sizes than PC and thus plays the micro-filling effect in the concrete to compact the microstructure. In the 80C + 20ZP concrete, these combined effects went against the dilution effect to make a dominant contribution to the 90-day strength enhancement.

### 3.2. SEM Analysis

SEM images of the 90-day cured concretes containing different amounts of ZP are shown in Figure 7. In the control concrete with 100% PC (Figure 7a), the formation of large amounts of hydration products indicates a relatively high hydration degree of the cement. In comparison, the concrete containing 20% ZP has a more compact microstructure because of the micro-filling effect and pozzolanic reaction of ZP (Figure 7b). This is consistent with its higher 90-day compressive strengths. When the ZP content increased to 40% and 60% (Figure 7c,d), some large capillary pores and cracks existed in the concretes, which are even interconnected to form a channel. The cross section is almost divided into two parts by the interconnected channel, indicating that the addition of 40% and 60% ZP degrades the microstructure.

### 3.3. Pore Structure

Figure 8 shows the effects of ZP addition on the pore structure of samples after 28 days of curing, and the relative parameters are tabulated in Table 3. The replacement of ZP for PC increases the total porosity of the samples. The development of a coarser pore structure was consistent with the strength reductions shown in Section 3.1. For the sample with 40% ZP content, the pore sizes were mainly concentrated in the range of <50 nm, which are mainly gel pores and small capillary pores. This suggests that more binder gel phase is formed in the matrix due to the hydration of ZP. In addition, more large capillary pores with diameters of >100 nm were formed in the sample with 60% ZP content. This was mainly due to the dilution effect obtained from the high volume of ZP addition.

Figure 9 shows the effects of ZP addition on the pore structure of samples after 90 days of curing, and the relative parameters are presented in Table 4. When compared with the 28-day cured samples, the extension of curing age effectively reduced the total porosity. The further hydration of PC and ZP particles produced more hydration products in the matrix to compact the pore structure. Moreover, the sample with 20% ZP content had a lower total porosity than the control sample. This demonstrated our previous speculation that after a long-term curing the reaction of 20% ZP is more evident to enhance the hydration degree of the matrix. It was consistent with the higher compressive strength observed in the 90-day cured concrete sample with 20% ZP.

### 3.4. Nanoindentation Investigation

Generally speaking, concrete matrix mainly consists of pore, hydration products and unhydrated particles. Figure 10 shows the typical load-penetration depth curves of different phases in the cement matrix of concrete samples after 90 days of curing. It can be found that the depths of different phases are not identical and the maximum depths of different phases are in the range of 135 to 775 nm, which depend on the nanomechanical properties of different phases. According to previous studies [27,30], the intrinsic elastic modulus of pore, low density C-S-H (LD C-S-H) gel, high density C-S-H (HD C-S-H) gel, CH and unhydrated particles was in the range of 0–12, 12–22, 22–34, 34–40 GPa and more than 40 GPa, respectively.

All nanoindentation results of three representative areas of the cement matrix were merged and the probability densities of the elastic modulus are shown in Figure 11. It can be observed that the probability densities of 12–40 GPa in all the concrete samples are maximum, and those of 0–12 GPa and more than 40 GPa are minimum. The indentation proportions of different phases based on the elastic modulus can be evaluated according to the frequency densities and the results can be considered to be equal to the volume percentages of different phases in the concrete samples, as shown in Table 5.

As shown in Figure 11 and Table 5, the volume percentage of C-S-H phase with the sum of LD C-S-H and HD C-S-H is the highest and that of pore is the lowest. Compared with the volume percentages of constituent phases of the control concrete sample, 20% ZP reduces the volume percentage of CH and pore, indicating that the higher pozzolanic reaction of ZP at the later stages consumed parts of CH to produce additional C-S-H to refine and fill the large pores. Furthermore, 20% ZP reduced the volume percentage of LD C-S-H but increased that of HD C-S-H, and the proportion of HD C-S-H in C-S-H was obviously increased. Therefore, the pozzolanic reaction of ZP significantly promotes the formation of HD C-S-H, which effectively increases the compressive strengths. Similarly, more than 20% ZP addition also increases the proportion of HD C-S-H, but decreases the total volume of C-S-H and increases the porosity of concrete samples. It may be due to the fact that more than 20% ZP will enhance the dilution effect, leading to a reduction in hydration products, and then resulting in concrete with lower strength. These results are in line with the SEM and MIP analyses shown in Section 3.2 and Section 3.3.

The ITZ of concretes containing different amounts of ZP were selected for nanoindentation investigation and the results are shown in Figure 12. The elastic modulus values of aggregate were almost more than 100 GPa and that of cement matrix were in the range of 25–45 GPa. For control concrete and concrete containing 20% ZP, the width of ITZ was around 20 µm. However, the addition of 40% and 60% ZP increased the widths of ITZ, which were about 30 µm and 40 µm, respectively. In addition, the average elastic modulus value of ITZ of concrete containing 20% ZP was 40 GPa, which was higher than that of control concrete (30 GPa). In comparison, the addition of 40% and 60% ZP could not improve the elastic modulus of ITZ. The 20% ZP addition could improve the nanomechanical properties of ITZ due to the filler effect and pozzolanic reaction of ZP. However, above a 20% critical amount, ZP addition enhanced the dilution effect and reduced the hydration degree to degrade the microstructure.

## 4. Conclusions

This work investigated the strength development and microstructure evolution of cement concretes containing ZP. The nanoindentation characterization was also used to evaluate the effects of ZP addition on the nanomechanical properties of concretes. The following conclusions can be drawn.

(1) The replacement of ZP in cement has a dilution effect on the hydration of concrete, resulting in a detrimental effect on strength development. Concretes with ZP addition generally exhibited lower compressive strengths than the control concrete with 100% PC. When the ZP content was 20%, the hydration of ZP went against the dilution effect and slightly enhanced the 90-day compressive strength.

(2) After 90 days of curing, the hydration of 20% ZP compacted the microstructure of the matrix. The MIP results showed that the sample with 20% ZP content had a lower total porosity than the pure PC sample. In comparison, some large capillary pores and cracks existed in concretes with higher ZP contents (40% and 60%). This increased the average pore sizes and total porosity of the samples.

(3) Nanoindentation investigation showed that the addition of 20% ZP decreased the porosity and the content of CH in the matrix, and also increased the contents of HD C-S-H. The addition of 20% ZP improved the nanomechanical properties of ITZ. However, a further increase in the content of ZP in the matrix mitigated the formation of C-S-H and increased the porosity to degrade the microstructure.

## Figures and Tables

**Figure 1 materials-13-04191-f001:**
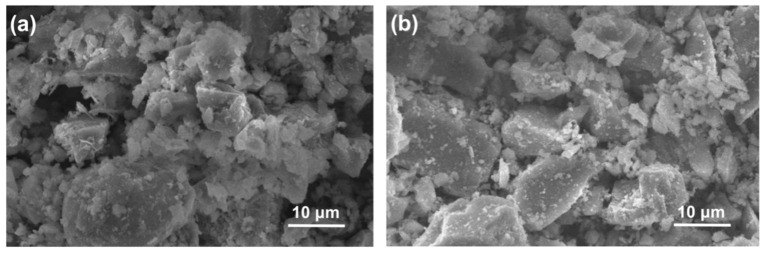
SEM images of (**a**) PC and (**b**) ZP.

**Figure 2 materials-13-04191-f002:**
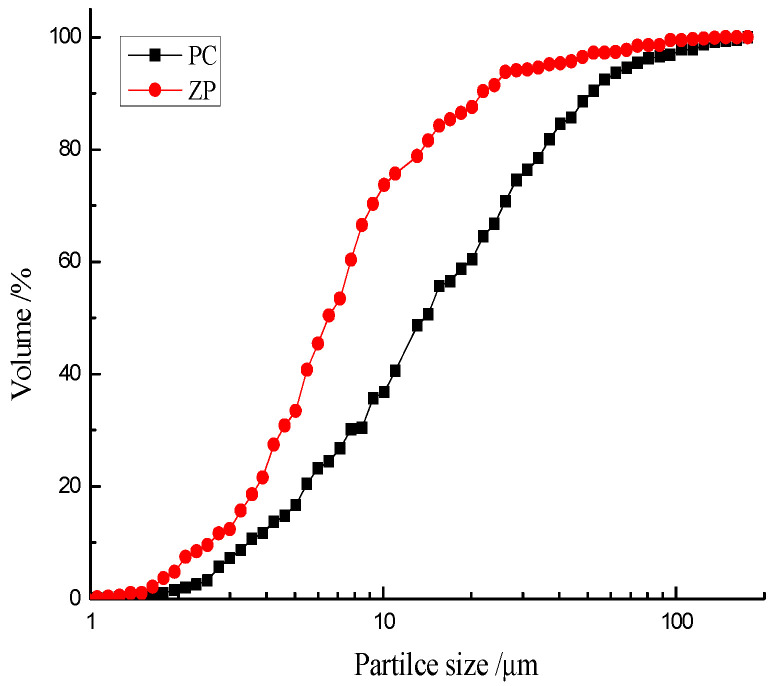
Particle size distribution curves of PC and ZP.

**Figure 3 materials-13-04191-f003:**
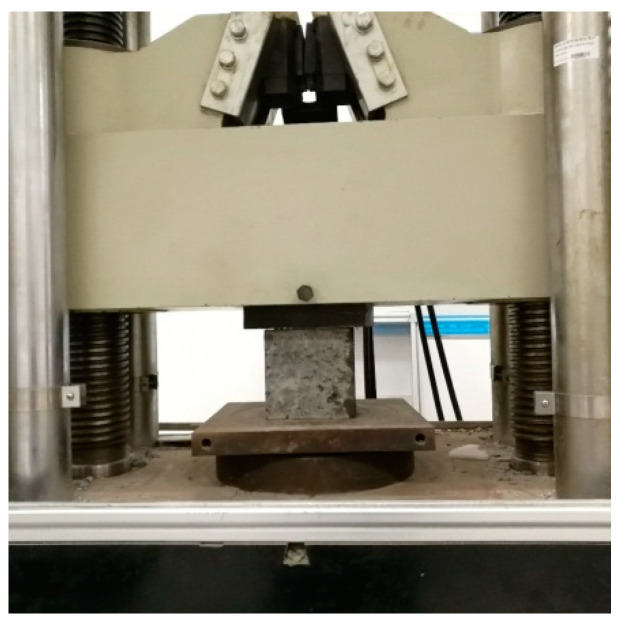
Compression test of concretes.

**Figure 4 materials-13-04191-f004:**
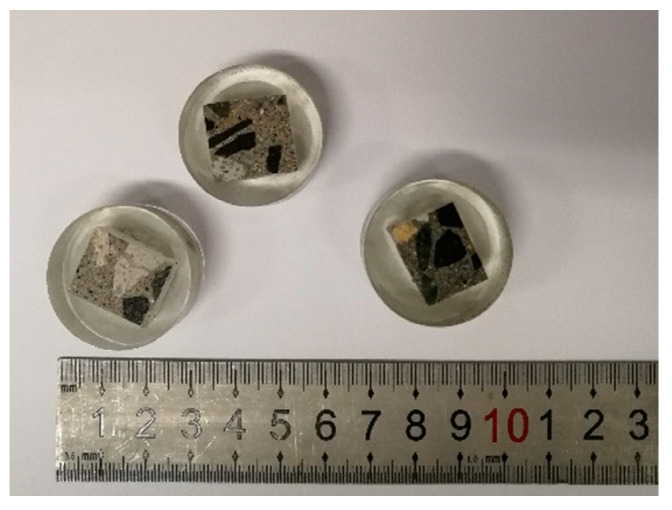
The samples for nanoindentation test.

**Figure 5 materials-13-04191-f005:**
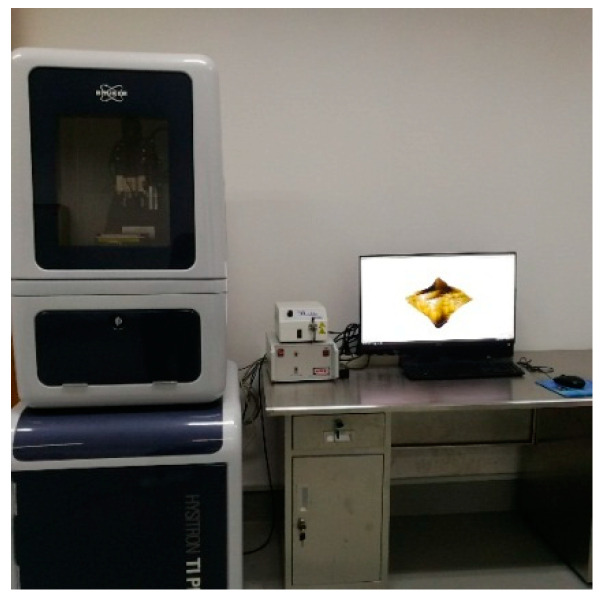
The instrument of nanoindentation.

**Figure 6 materials-13-04191-f006:**
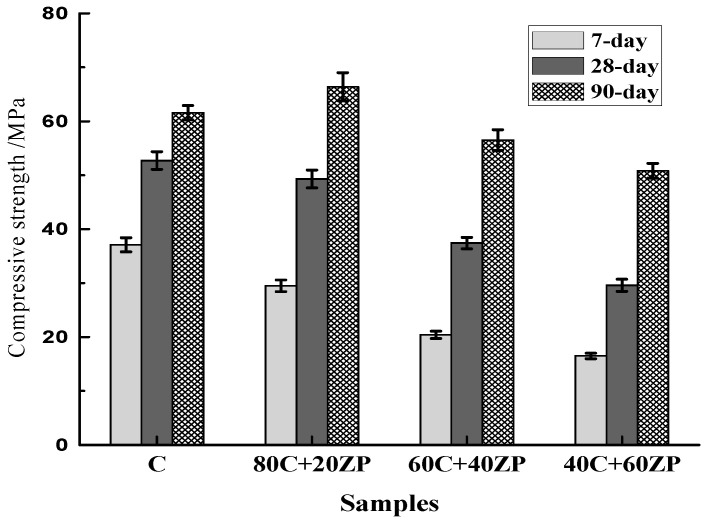
Compressive strengths of concrete samples.

**Figure 7 materials-13-04191-f007:**
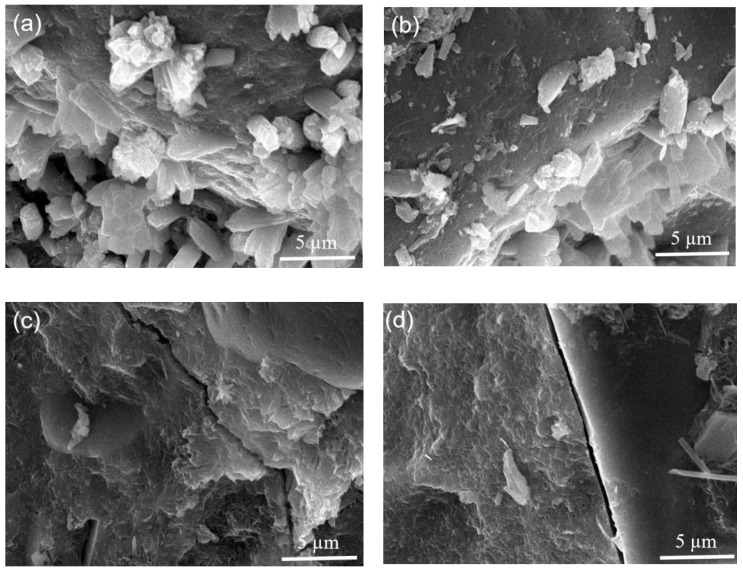
SEM images of (**a**) C, (**b**) 80C + 20ZP, (**c**) 60C + 40ZP and (**d**) 40C + 60ZP.

**Figure 8 materials-13-04191-f008:**
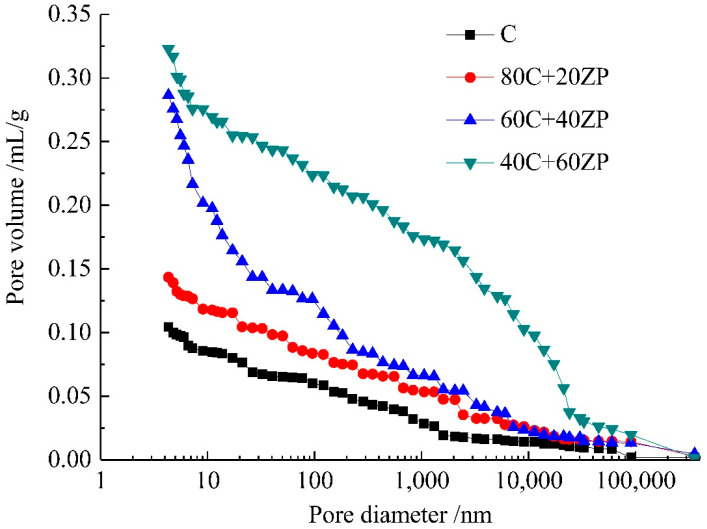
Pore size distribution curves of 28-day cured samples.

**Figure 9 materials-13-04191-f009:**
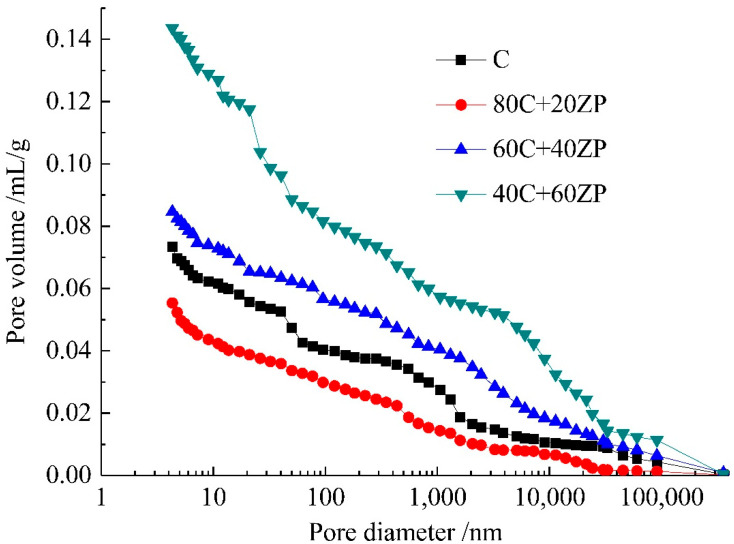
Pore size distribution curves of 90-day cured samples.

**Figure 10 materials-13-04191-f010:**
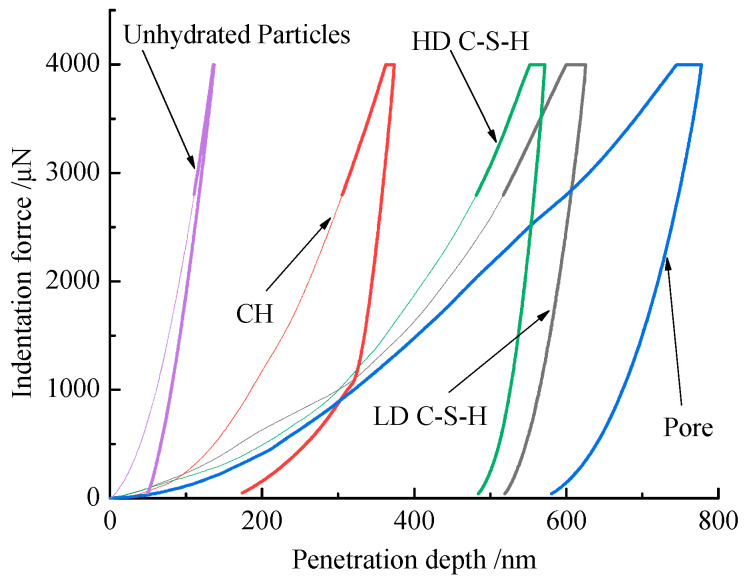
Typical load-penetration depth curves in concrete matrix.

**Figure 11 materials-13-04191-f011:**
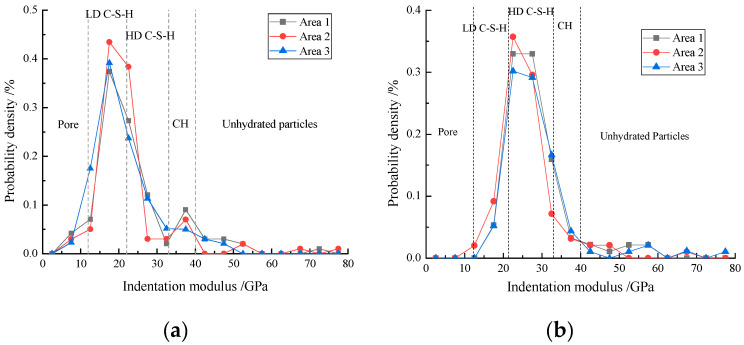
Probability density of elastic modulus of (**a**) C, (**b**) 80C + 20ZP, (**c**) 60C + 40ZP and (**d**) 40C + 60ZP.

**Figure 12 materials-13-04191-f012:**
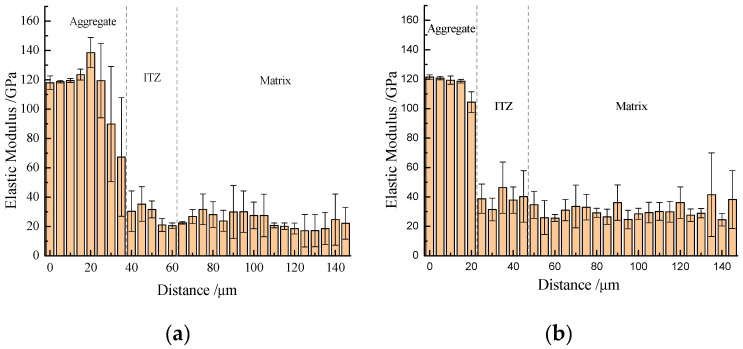
Nanoindentation modulus on ITZ of (**a**) C, (**b**) 80C + 20ZP, (**c**) 60C + 40ZP and (**d**) 40C + 60ZP.

**Table 1 materials-13-04191-t001:** Chemical compositions of PC and ZP (wt.%).

Materials	SO_3_	SiO_2_	Fe_2_O_3_	Al_2_O_3_	CaO	MgO	K_2_O	LOI
PC	2.11	23.12	2.88	4.12	61.68	0.98	0.21	3.23
ZP	0.71	66.98	1.21	19.88	3.67	0.49	0.06	4.71

**Table 2 materials-13-04191-t002:** Mix proportions of concretes (kg/m^3^).

Samples	Cement	ZP	River Sand	Crushed Granite	Water	Superplasticizer
C	450	0	726	1089	135	3.05
80C + 20ZP	360	90	726	1089	135	3.74
60C + 40ZP	270	180	726	1089	135	4.32
40C + 60ZP	180	270	726	1089	135	4.85

**Table 3 materials-13-04191-t003:** Pore structure parameters of the 28-day cured samples.

Sample	<50 nm (mL/g)	50~100 nm (mL/g)	>100 nm (mL/g)	Total Porosity (mL/g)
C	0.039	0.005	0.060	0.104
80C + 20ZP	0.046	0.014	0.084	0.143
60C + 40ZP	0.153	0.007	0.126	0.286
40C + 60ZP	0.080	0.020	0.224	0.323

**Table 4 materials-13-04191-t004:** Pore structure parameters of the 90-day cured samples.

Sample	<50 nm (mL/g)	50~100 nm (mL/g)	>100 nm (mL/g)	Total Porosity (mL/g)
C	0.026	0.007	0.040	0.073
80C + 20ZP	0.022	0.004	0.030	0.055
60C + 40ZP	0.022	0.006	0.057	0.085
40C + 60ZP	0.055	0.007	0.082	0.144

**Table 5 materials-13-04191-t005:** Volume percentages of each phase in the cement matrix (%).

Samples	Pore	LD C-S-H Gel	HD C-S-H Gel	CH	Unhydrated Particles
C	4.21	51.14	23.42	12.25	7.21
80C + 20ZP	3.34	23.41	53.25	8.31	8.95
60C + 40ZP	5.21	18.32	50.21	7.12	15.42
40C + 60ZP	6.54	16.54	41.45	4.52	23.12

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
