# Peer review of "Strengths, Microstructure and Nanomechanical Properties of Concrete Containing High Volume of Zeolite Powder"

_materials, 2020, doi:10.3390/ma13184191_

Round 1

Reviewer 1 Report

There are some weaknesses through the manuscript which need improvement. Therefore, the submitted manuscript cannot be accepted for publication in this form, but it has a chance of acceptance after a major revision. My comments and suggestions are as follows:

1- Abstract gives information on the main feature of the performed study, but some details about the examined concrete must be added. However, a concise abstract is needed.

2- Authors must clarify necessity of the performed research. Aims of the study, and also differences with the previous researches must be clearly mentioned.

3- The literature study must be enriched. For instance, authors must refer to the methods used to determine mechanical properties of reinforced concrete in the following papers: a. Adv. Eng. Software (2018) 127:51-58 (doi.org/ 10.1016/j.advengsoft.2018.10.002) and b. Constr. Build. Mater. (2020) 261:119962 (doi.org/ 10.1016/j.conbuildmat.2020.119962)

4- There are different methods and materials to reinforce concrete. Author must explain why zeolite powder was selected.  

5- As the manuscript deals with different mixtures, mixtures in previous studies must be compared. For example, authors must refer to https://doi.org/10.1016/j.engstruct.2019.109844.

6- The manuscript presented experimental tests. For instance, compression tests on the concrete sample. Therefore, it is necessary to illustrate some figures (not schematic) to show experimental setup and specimens under test conditions.

7- Since experimental tests (e.g., compression, nanoindentation, …) were conducted, details of experimental practice must be presented. Relevant studies must be cited.

8- In its language layer, the manuscript should be considered for English language editing. There are sentences which have to be rewritten.

9- The conclusion must be more than just a summary of the manuscript. List of references must be updated based on the proposed papers. Please provide all changes by red color in the revised version.

Reviewer 2 Report

A careful review of the manuscript “Strengths, microstructure and nanomechanical properties of concrete containing high volume of zeolite powder“ has been completed. Despite the fact that the authors discuss the effects of ZP on the microstructure evolution of concrete, it is not really clear what is the main objective of the manuscript and how these systems can be implemented in practice; also it is very difficult to follow what is presented. This paper is useful for engineers to see closely to ZP dilution effect on the concrete. However, this investigation is not comprehensive and there are still rooms to improve. Therefore, this manuscript is not recommended for publication in Journal of materials due to the fact that paper has a critical and serious problem explained below:

  1. English needs to be improved. I had difficulty to follow the text and had to read the same sentence several times.
  2. The originality is not explained in detail.
  3. References are not cited sufficiently and appropriately.
  4. Please provide more detailed definitions on the Results and discussion used in this paper.

I recommend that the paper is rejected in its current form and a request is made to revise and re-submit for review.

Reviewer 3 Report

Dear Authors,

The paper Strengths, microstructure and nanomechanical properties of concrete containing high volume of zeolite powder by Zhouping Yu, Weijun Yang, Peimin Zhan, Xian Liu and Deng Chen is well suited for journal Materials. The authors of this article analyzed the results of the present studies on effects of zeolite powder (ZP) on evolution of the concrete microstructure and mechanical properties after 7, 28, 90 days of curing.

The paper is scientifically valuable. The article contains chapters in the order typical for research articles: introduction, experimental (materials, mixture proportions), results and discussion, conclusions.

The title in the reviewer opinion accurately reflect the content.

The information contained in the Abstract interest readers to read the article, but, in the reviewer opinion, does not fully describe the content of the article. Should be added information that the tests included samples with 20, 40, 60ZP (it should be indicated that the values do not represent "%" in relation to cement).

Introduction present background of analyzed problem, literature review shows important achievements of earlier articles.  Many articles in the literature have been published in the last 3 years, although none of the journal Materials. At the end of chapter 1, part of the description of the microstructures of the concrete samples study has been replaced with a reference to [22], which should be considered appropriate.

In chapter 2, table 2, first column - the reviewer recommends considering the possibility of changing the descriptions, e.g. instead of "C + 60ZP" maybe "40C + 60ZP" would be better.

The results of conducted tests (chapter 3) complement the known results and give answers in terms of higher additive content ZP, its influence on the structure and strength of concrete samples after 7, 28 and 90 days. In the reviewer opinion, the authors should consider whether it is possible to use concretes with a higher ZP in special applications.

The summary is too short, the previously given conclusions should be mentioned. The summary gives an explicit reference to 20 ZP. Other values are just background. The proportions should be changed. The research is for several ZP values, conclusions should be structured in such a way as to give "tendencies across all values" and indicate 20ZP as the most favorable value, additionally giving reasons why this value is the most favorable.

The article was written enough well in English, is understandable for a reviewer, a person who does not speak English as a mother tongue.

Reviewer 4 Report

The paper topic is good and the paper is well written.  The following needs to be clarified.

Line 35 and 36: et al. is commonly used in citations not materials

Line 69: Change "Published the literature" to "published literature"

Line 91: show that ZP is added as cement replacement

Line 100-111: Using a previously broken sample under SEM might affect the results. Was anything done to avoid this?

Round 2

Reviewer 1 Report

Dear Authors,

you have addressed the comments, and answered the questions. The revised version of your manuscript appears to be suitable for publication.

Reviewer 2 Report

Ok